# Effects of Three-Directional Seismic Wave on Dynamic Response and Failure Behavior of High-Steep Rock Slide

**Ziwei Ge and Hongyan Liu \***

School of Engineering and Technology, China University of Geosciences (Beijing), Beijing 100083, China; geziwei@cugb.edu.cn
\* Correspondence: lhy1204@cugb.edu.cn

**Abstract:** The landslide triggered by earthquakes can cause severe infrastructure losses or even fatalities. The high-steep rock slide is the most common type of landslide in the earthquake area. In an earthquake, the ground moves randomly in all directions, two horizontal directions (East-West (EW) direction, North-South (NS) direction) and one vertical direction (Up-Down (UD) direction). Even though extensive studies have been carried out on the earthquake-triggered landslide, the effects of each single seismic wave and the three-directional seismic waves are not considered. This study aims to evaluate the effects of different types of the seismic waves on the dynamic response and failure behavior of the high-steep rock slide. To investigate the effects of each single seismic wave and three-directional seismic wave, this study presents a numerical model with four types of seismic waves, e.g., East-West (EW) direction, North-South (NS) direction, Up-Down (UD) direction, and three-directional wave (EW_NS_UD). The numerical results revealed that the types of the seismic waves have significantly different effects on the dynamic process, failure behavior, run-out distance, velocity, and deposition of the high-steep rock slide.

**Keywords:** three-directional seismic wave; rock slide; dynamic response; failure behavior; numerical modeling





## 1. Introduction

The earthquake-triggered landslide represents a dangerous natural hazard causing significant damage to property and infrastructure and personal casualties [1,2]. There have been many reports on the earthquake-triggered landslides in the past few years [3–5]. For example, on 8 October 2005, the Kashmir earthquake triggered a large number of landslides in the northern Pakistan. The most disastrous landslide destroyed three villages and caused nearly 1000 deaths [6]. A total of 60,104 landslides induced by the 2008 Wenchuan earthquake (Ms = 8.0) directly killed over 20,000 people [7]. Less than five years later, the 2017 Jiuzhaigou earthquake shocked the Sichuan province, China, and induced as many as 179 landslides [8,9]. In spite of their significant damage to property and human casualties, the cause and effect of earthquake-triggered landslides is not well understood [10,11].

The process of earthquake-triggered landslides is complex, which is also called the dynamic response of the landslide [12]. When the seismic wave reaches the ground surface, the dynamic behavior of the seismic wave will be affected and modified by the soil. Thus, it appears possible to reduce the seismic response by intervening and modifying the characteristics of the soil [13–15]. The failure behavior of the earthquake-triggered landslide is also closely related to its dynamic response under the seismic waves. The earliest study on the effect of the seismic waves can be traced back to 1965, when Newmark proposed a method permitting a rapid estimate of the dams and embankments displacement in an earthquake [16]. After that, many studies focusing on the effects of seismic waves on the earthquake-triggered landslide dynamic response and failure behavior have been published [17–21]. For example, Chuang et al. developed a nowcasting model for estimating the probability of landslides induced by earthquakes [22]. Zou et al. analyzed

the controlling factors of the distribution of the coseismic landslides [23]. Valagussa et al. investigated that larger landslides were associated with higher ground motion and explained why stronger shaking could induce larger landslides [24]. Chen et al. modeled a symmetrical slope by using the DDA (discontinuous deformation analysis) method, and the results showed that pulse-like ground motion (PLGM) might be the main factor for the earthquake-triggered landslides [25].

In spite of the extensive studies on the earthquake-triggered landslides, how to consider the effect of the seismic wave on the landslide dynamic process is not well understood. 3DEC (the three-dimensional discrete element code) is a powerful tool in numerically modeling the dynamic response and failure behavior of the earthquake-triggered landslides [26]. The input of the seismic wave in 3DEC mainly depends on the seismic wave data measured in the field. However, in an earthquake, the ground moves randomly in all directions, both horizontally and vertically. The earthquake accelerations are available for motion in two horizontal directions (East-West (EW) direction and North-South (NS) direction) as well as in the vertical direction (Up-Down (UD) direction). Therefore, researchers have not reached the same conclusion on which one of the seismic waves should be selected as the input of the seismic wave [27,28]. For example, Gazetas et al. assumed that the vertical seismic wave has no apparent effect on landslides [29]. However, Sun et al. analyzed the effect of the vertical seismic waves on the failure mechanism of a practical earthquake-triggered rock avalanche and concluded that the vertical seismic force played an important role. Cui et al. obtained that the horizontal acceleration from P-waves played a dominant role in the failure of the slope with the discrete element method [30]. Zhang et al. used the DDA method to simulate the failure process of the Daguangbao landslide under the action of multidirectional seismic waves. The results showed that the simulation results under the combined action of transverse and longitudinal waves were more consistent with the actual failure behavior [31].

Moreover, the effect of the three-directional seismic waves on the slope failure is rarely considered in the existing studies. To address the above problems, in this study, we consider the three-directional seismic waves and focus on the effect of different seismic waves on the dynamic response and failure behavior of the landslide. A typical high-steep rock slide numerical model with four cases is adopted to simulate the effects of seismic waves on the landslide mobility characteristics with 3DEC. This study aims to evaluate the effects of the different seismic waves on the dynamic response and failure behavior of the high-steep rock slide.

This rest of this paper is organized as follows. Section 2 describes the characterizations of the Jiuzhaigou earthquake and the high-steep rock slide. Section 3 presents the building of the high-steep rock slide numerical model, the determination of the numerical parameters, and the processing of the seismic wave loadings. Section 4 lists the numerical results and the analysis of the numerical results. Section 5 introduces the discussions in the numerical investigation. Section 6 draws the conclusions in this study.

## 2. Study Area: The Jiuzhaigou Earthquake and the High-Steep Rock Slide

### 2.1. The Jiuzhaigou Earthquake

The earthquake with a magnitude of Ms = 7.0 occurred in Jiuzhaigou County, Sichuan province, China, at 21:19 CST on 8 August 2017. The epicenter was located in the world natural heritage Jiuzhaigou scenic area (33.20° N, 103.82° E), Zhangzha Town, 39 km east of Jiuzhaigou County, 285 km south of Chengdu. The focal depth of the earthquake was 20 km, and the maximum intensity was IX degrees. The long axis of the isoseismal lines is generally north-northwest. The total area of the VI degrees or above was 18,295 km$^2$, which caused disasters in eight counties in the Sichuan and Gansu provinces. The earthquake caused 25 deaths, injured 525 people, caused six people to become lost, and damaged 73,671 houses.

The Jiuzhaigou area is located at the junction of the Danba-Wenchuan, Songpan-Ganzi and the Motianling in the West Qinling tectonic zone (Figure 1). Bounded by the Tazang

fault (TZF), the Xueshan fault (XSF), and the Minjiang fault (MJF), the north-east and the north-west corners are the Songpan-Ganzi orogenic zone, the Animaqing orogenic zone (anticline), and the Malkang thrust-detachment rock slice (syncline). The middle part of the Jiuzhaigou area is the Motianling nappe in the west orogenic zone. The Minjiang fault (MJF) is located in the west side of the epicenter. The overall trend is north-south, and the section is inclined to the west. It shows upthrust and left-lateral strike-slip action, indicating that the Minjiang fault (MJF) is not the seismogenic fault of the earthquake. However, it has a certain restrictive effect on the aftershocks, surface, and the western boundary of the earthquake landslide distribution of the earthquake. The Tazang fault (TZF) is located in the north side of the epicenter. The overall trend is northwest-southeast, and the fracture dip angle is 50°–60°. It also had a specific limit on the aftershocks of the earthquake, surface deformation, and the northern boundary of the distribution of the earthquake area. Therefore, it can be inferred that the seismogenic fault of this earthquake was the Huya fault (HYF).

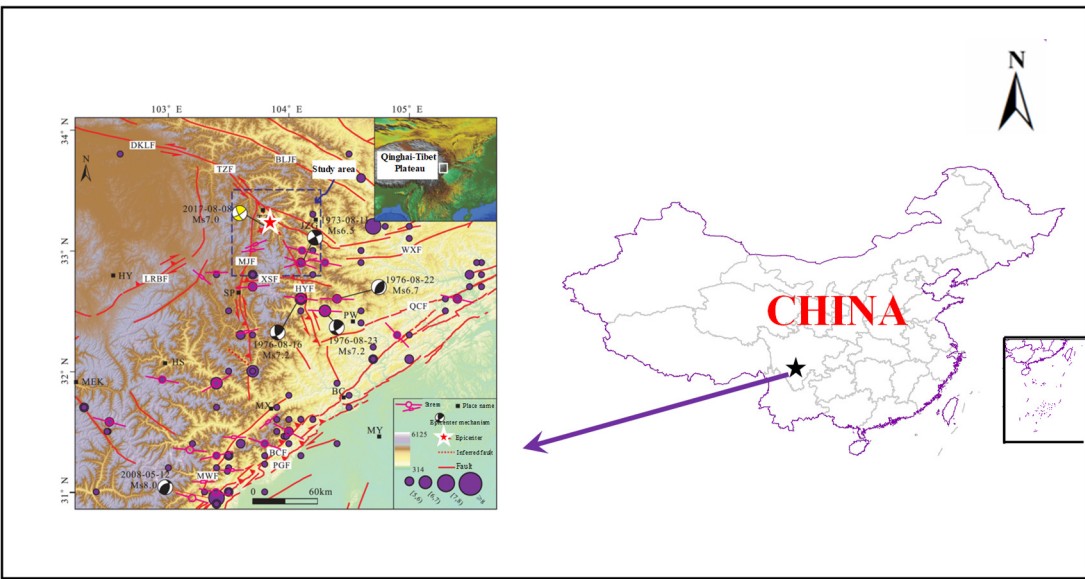

**Figure 1.** Geology of the Jiuzhaigou area and the location of the Jiuzhaigou earthquake.

Fault: DKLF, Dongkunlun fault; TZF, Tazang fault; BLJF, Bailongjiang fault; WXF, Wenxian fault; LBF, Longriba fault; MJF, Minjiang fault; HYF, Huya fault; XSF, Xueshan fault; QCF, Qingchuan fault; MWF, Maowen fault; BCF, Beichuan fault; PGF, Pengguan fault.

Place names: TZ, Tazang; JZG, Jiuzhaigou; HY, Hongyuan; SP, Songpan; PW, Pingwu; HS, Heishui; MEK, Malkang; MX, Maoxian; BC, Beichuan; MY, Mianyang

## 2.2. The High-Steep Rock Slide

The high-steep slope is one of the most common typical landforms in mountainous areas, which refers to the slope with an approximately vertical angle. The high-steep slope is generally composed of horizontal sedimentary rocks. Due to its steep terrain characteristic, rock slides often develop on the surface of the high-steep slope. The high-steep rock slide is one of the most common type of the landslide in the Jiuzhaigou area. For example, Figure 2a shows the perilous rock mass at a high-steep slope on the right bank of the Swan-sea central highway (33.09° N, 103.66° E). Figure 2b shows the overall view of the rock slide at multiple points on the right bank of the Swan-sea central highway.

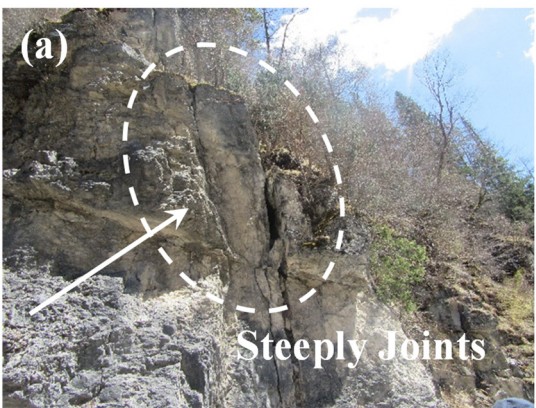 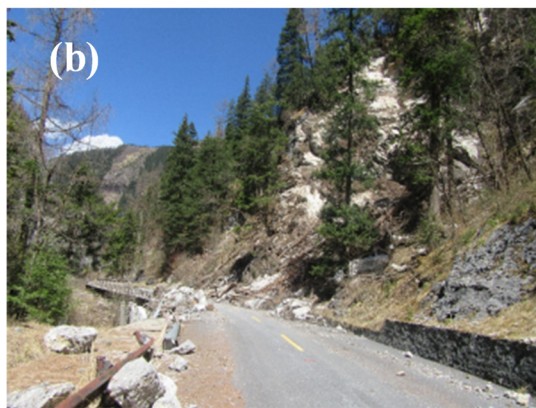

**Figure 2.** The high-steep slope (**a**) and the general view of the high-steep rock slide on the right bank of the Swan-sea central highway (**b**).

The perilous rock mass on the surface of the high-steep slope is cut by vertical tensile joints or steeply dipping columnar joints [32]. It separates from the stable base rock and forms tall, narrow, upright, laminated rock columns. Finally, the rock columns are divided into blocks by two sets of nearly orthogonal structural planes, as shown in Figure 3. The strata is the general term for all stratified rocks. The high-steep slope is most prone to rock slide. Under the action of the earthquake, the perilous rock mass suddenly separates from the base rock. The stability of the high-steep rock slide is mainly affected by the overturning moment, as shown in Figure 4. When the overturning moment is greater than the anti-titling moment, a toppling failure occurs, and then it will continue to collapse, roll, fall, etc.

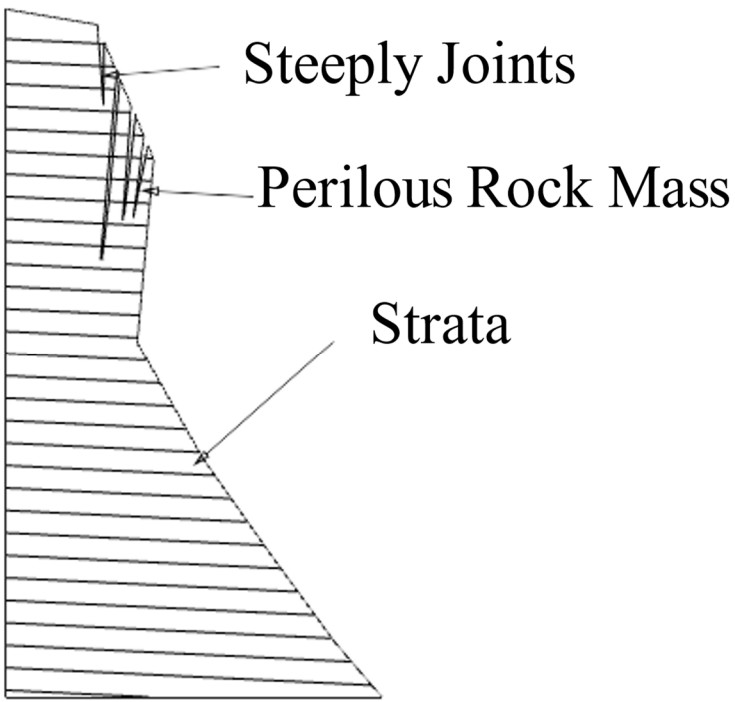

**Figure 3.** Typical forward strata layer of high-steep slope.

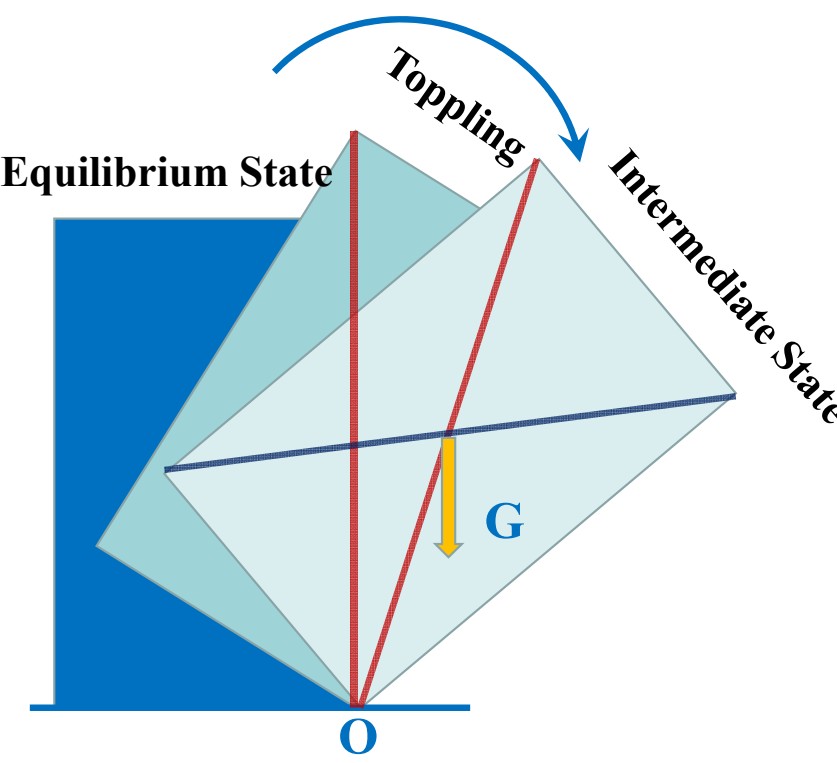

**Figure 4.** The overturning moment of the high-steep rock slide.

### 3. Method: Numerical Investigations of Effects of Seismic Wave on the Dynamic Response and Failure Behavior

*3.1. Overview*

The process of the landslide failure involves the separation of the blocks and the reconstruction of the contact surface. Therefore, the numerical method requires that it allows the blocks can move freely. The discrete element method (DEM) is a numerical method especially suitable for stress analysis of jointed rock masses [33]. The key concept of DEM is based on Newton's law, which treats the research object as an assemblage of the rigid or deformable blocks, and the location of each block is updated during the entire deformation/motion process [34]. Therefore, the DEM is widely used in the process of the dynamic response and failure behavior of the high-steep rock slide. The DEM program called 3DEC is used for numerical modeling.

To investigate effects of the dynamic response and failure behavior of the high-steep rock slide under different seismic wavs, a typical high-steep rock slide numerical model is built in 3DEC. As illustrated in Figure 5, the research process is divided into three main parts: preprocessing of the numerical modeling (steps 1–2), numerical modeling (steps 3–6), and analysis of the numerical results (step 7). Specifically, the research steps are (1) filtering and baseline correction of the seismic waves; (2) calculation of the appropriate mesh size; (3) building of the high-steep rock slide; (4) determination of the numerical parameters; (5) inputting of the processed seismic waves; (6) setting of the monitors of the key blocks; and (7) analyzing of the numerical results.

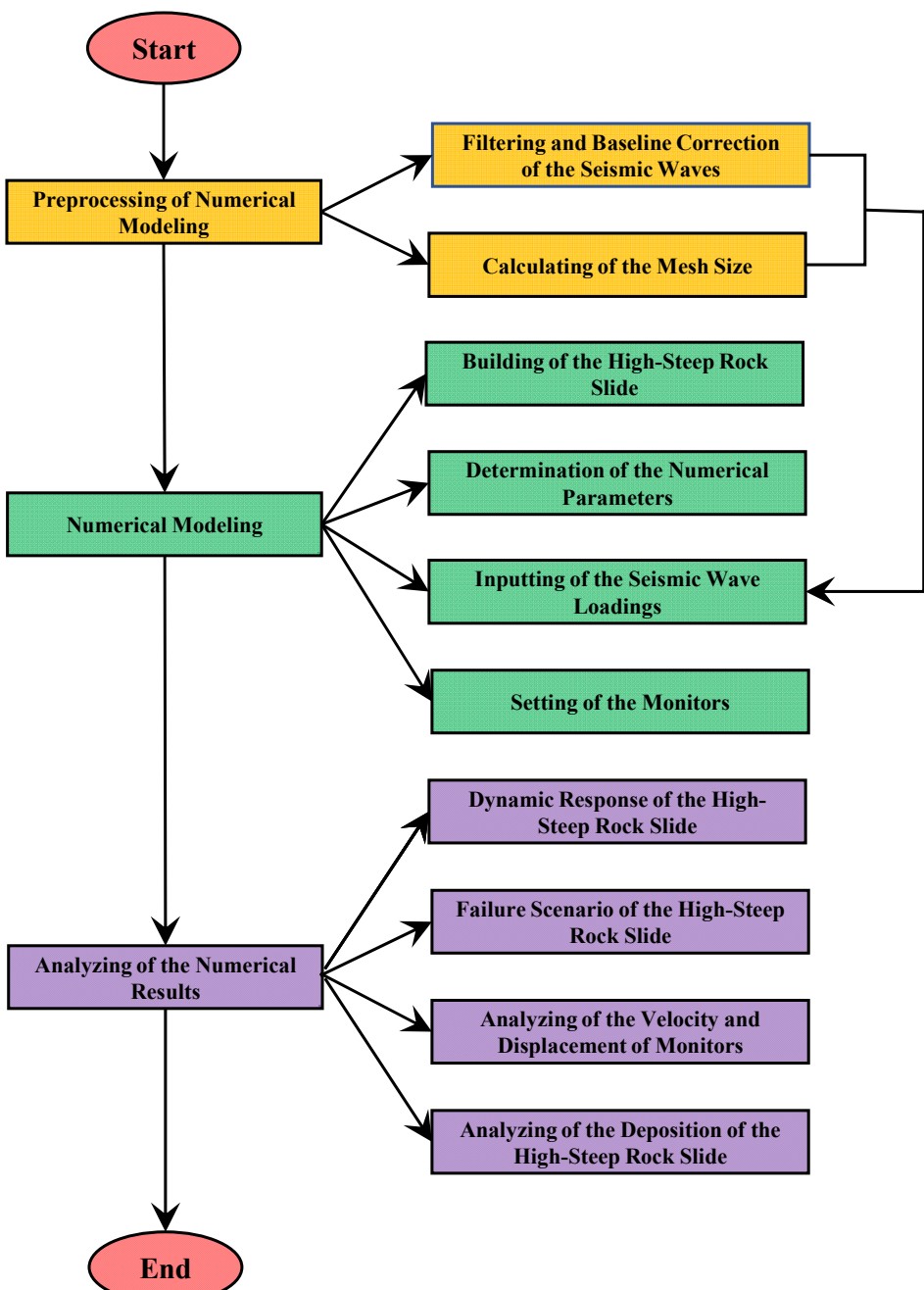

**Figure 5.** The flowchart of numerical investigation on the effects of the dynamic response and failure behavior of the high-steep rock slide.

### 3.2. Details of Dynamic Modeling Considerations in the Discrete Element Method

There are three issues should be considered when preparing a DEM model for a dynamic analysis, e.g., boundary conditions, local damping, and max unbalanced force. The boundary conditions are essential in 3DEC dynamic analysis. Figure 6 shows the setting of the slope boundary conditions, in which the viscous boundary is to set up normal and tangential viscous dampers on the boundary of the model. The normal and tangential viscous dampers can provide unbalanced force to absorb external incident waves. The

unbalanced force will be applied to the viscous boundary, and the forces in the three coordinate directions are as follows:

$$
\begin{aligned}
F_x &= -\rho C_{\mathrm{p}}\left(v_x^m - v_x^{ff}\right)A + F_x^{ff} \\
F_y &= -\rho C_{\mathrm{s}}\left(v_y^m - v_y^{ff}\right)A + F_y^{ff} \\
F_z &= -\rho C_{\mathrm{s}}\left(v_z^m - v_z^{ff}\right)A + F_z^{ff}
\end{aligned}
\tag{1}
$$

where, $\rho$ is the mass density; $C_{\mathrm{s}}$ and $C_{\mathrm{p}}$ are the velocities of longitudinal and transverse waves; and $A$ is the area affected by the viscous boundary grid. $v_x^m$, $v_y^m$, $v_z^m / v_x^{ff}$, $v_y^{ff}$, and $v_z^{ff}$ are the three direction velocity components of the lateral boundary nodes of the main grid/viscous boundary network, respectively. $F_x^{ff}$, $F_y^{ff}$, and $F_z^{ff}$ are the forces on the viscous boundary grid. Therefore, the wave propagation will not be affected by the distortion caused by the boundary conditions.

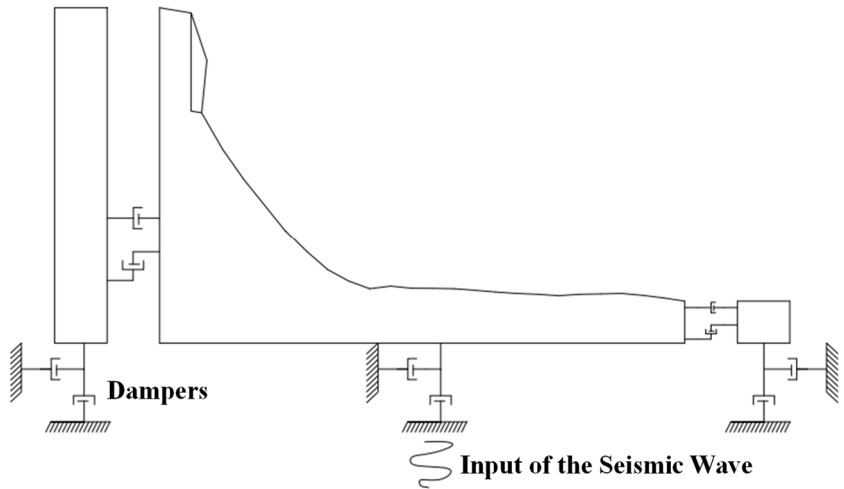

**Figure 6.** Dynamic viscous boundary condition.

The local damping is used in the 3DEC dynamic analysis to reduce the unnecessary numerical vibrations block motion system. In general, the vibration amplitude of any vibration system will decrease due to air resistance, friction, etc. However, in the 3DEC dynamic analysis, the kinetic energy cannot be dissipated naturally. Therefore, the appropriate local damping is added to ensure the normal vibrations of the numerical model. It takes many repeated trials to obtain an appropriate value. In this study, the local damping ratio is set to 0.005.

The maximum unbalanced force is monitored to detect whether the system has reached a state of equilibrium. The ideal model balance means that the nodal force vector at the center of the block is exactly zero. However, the maximum unbalanced force in numerical analysis can never reach zero. Generally, it is considered that the maximum unbalanced force is sufficiently small compared with the representative force of the model. It represents the numerical modeling has reached a state of equilibrium. The maximum unbalanced force in this study is set to 0.00002.

*3.3. Building of the High-Steep Rock Slide Numerical Model*

As shown in Figure 2a, the rock slide is located in Jiuzhaigou with an altitude of 320 m, length of 200 m, and slope angle of about 50°, where a 10 m × 90 m × 20 m dangerous rock developed at an altitude of 220 m. The strata in the rock slide area are mainly exposed to the limestone of the second member of Daguanshan formation (Cpd) of carbon permian, with strong weathering and well-developed joints. The strata occurrences are 90°∠20° and

4-m spacing, and the filling is mudstone. The slope ends at about 200 m, followed by a ~10° and ~300-m open terrain.

To facilitate the study of the effect of three-directional seismic waves on the high-steep rock slide, the numerical model is simplified comparing with the actual situation. Figure 7a shows the numerical model according to Figure 2a. The numerical modeling adopts the left-handed coordinate system. In the discrete element analysis, multiple sets of horizontal and longitudinal structural planes are often used to form discrete blocks so that each block can move freely. One joint set can divide the perilous rock into planes, while two joint sets can separate the perilous into columns. Therefore, we need three joint sets to cut the rock into blocks. Three sets of discontinuities cut the perilous rock mass, creating an assemblage of blocks with a spacing of 4 m. Three joints set have dip direction/dip angles of 90°∠20°, 0°∠90°, and 90°∠90°. Considering the lateral movement of the landslide, the size of the base rock is 200 m in width, which is much larger than the 20-m width perilous rock mass. Furthermore, seven monitors are set to explain the ground motion response of the failure process and the movement characteristics clearly, as shown in Figure 7b.

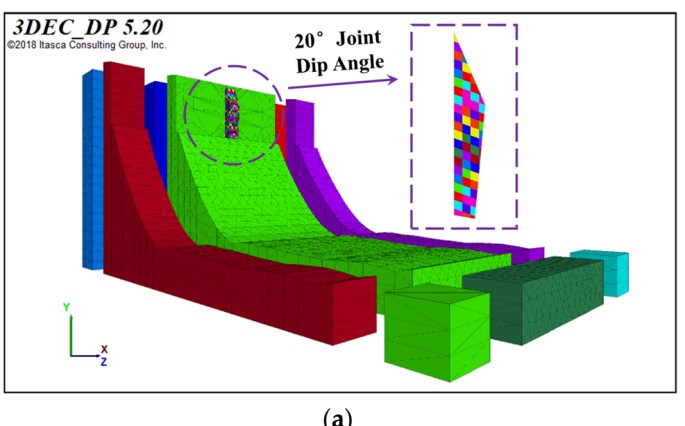
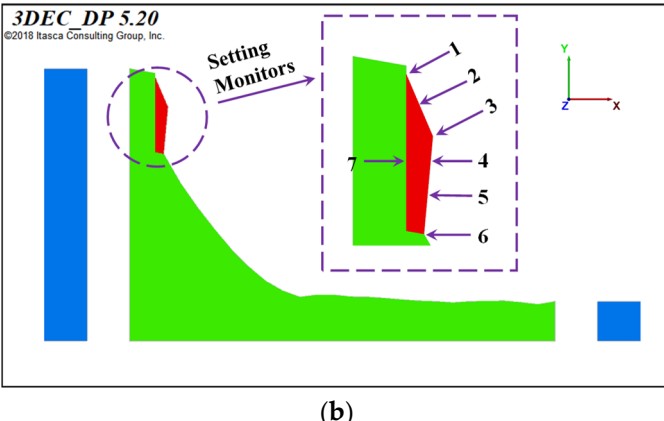

(**a**)                                    (**b**)

**Figure 7.** Numerical model of the high-steep slope in 3DEC (**a**) and the locations of the monitors (**b**).

### 3.4. Determination of the Numerical Parameters

Table 1 illustrates the physical and mechanical parameters for the numerical analysis [26]. All blocks and joints follow the linearly elastic model and Coulomb-slip model, respectively. It is assumed that the numerical model is in a residual state because only the parameters after slope failure can be obtained.

**Table 1.** Physical and mechanical parameters for the numerical analysis.

| Item | Value | |
|---|---|---|
| Fixed base rock (limestones) | Density (kg·m$^{-3}$) | 2950 |
| | Bulk modulus (GPa) | 36.63 |
| | Shear modulus (GPa) | 16.69 |
| | Friction angle (°) | 44 |
| | Cohesion (MPa) | 10.32 |
| Sliding rock (sandstones) | Density (kg·m$^{-3}$) | 2860 |
| | Bulk modulus (GPa) | 25.28 |
| | Shear modulus (GPa) | 13.84 |
| | Friction angle (°) | 36 |
| | Cohesion (MPa) | 7.84 |
| Joint (mudstone filling) | Normal stiffness (MPa·m$^{-1}$) | 20 |
| | Shear stiffness (MPa·m$^{-1}$) | 15.2 |
| | Friction angle (°) | 32.1 |
| | Cohesion (MPa) | 0.9 |
| Sliding surface (sandstone contact) | Normal stiffness (MPa·m$^{-1}$) | 7.2 |
| | Shear stiffness (MPa·m$^{-1}$) | 4.8 |
| | Friction angle (°) | 27.8 |
| | Cohesion (MPa) | 0.4 |

### 3.5. Processing of the Input Seismic Wave Loadings

Figure 8 shows the original EW (East-West direction), NS (North-South direction), and UD (Up-Down direction) seismic waves used in the numerical model. These curves represent the acceleration time histories of the seismic waves. They are obtained from the 2017 Jiuzhaigou earthquake, corresponding to the *X*, *Z*, and *Y* directions of the dynamic load input during the simulation process. The geographic coordinates of the selected seismic wave station are 32.8° N 105.4° E. The dataset is provided by the China Earthquake Networks Center, National Earthquake Data Center (http://data.earthquake.cn (accessed on 20 March 2020)).

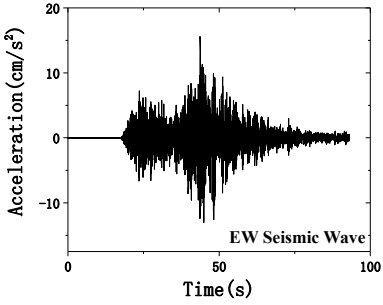 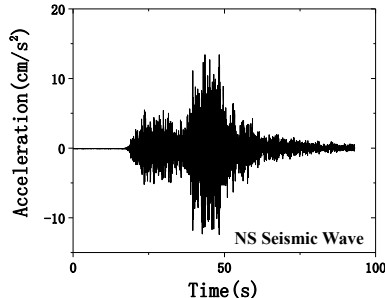 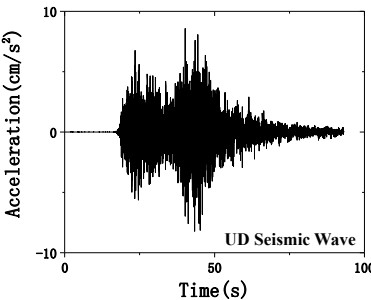

**Figure 8.** Ground acceleration records of Jiuzhaigou earthquake.

The frequency component of the input waveform, and the wave velocity characteristics of the rock mass will affect the numerical accuracy of wave propagation. To describe the wave propagation in the model accurately, the study by Kuhlemeye and Lysmer shows that the zone size must be less than 1/8~1/10 of the wavelength, corresponding to the highest frequency of the input waveform [35]. On account of the maximum frequency of seismic waves has a more significant impact on the zone size, filtering seismic waves is required. The higher the maximum frequency is, the smaller the zone size is. The aim of filtering seismic wave is to filter out the high-frequency components of the original waveform and reduce the maximum frequency of small seismic waves, thereby increasing the minimum zone size for less calculation time.

As an illustration, Figure 9a shows the seismic wave filtering of the EW seismic wave. The frequency corresponding to peak acceleration is mainly concentrated in the range of 10~15 Hz. Therefore, the part of the waveform greater than 20 Hz can be filtered by Wizard software. The maximum frequency of the filtered waveform is about 20 Hz, as shown in the red line in Figure 9a. The equation for calculating the shear wave velocity is $C_s = \sqrt{G/\rho}$, and the equation for calculating the longitudinal wave velocity is $C_p = \sqrt{\frac{K+4G/3}{\rho}}$, where $G$, $K$ and $\rho$ represent the model shear modulus, bulk modulus, and density, respectively. According to the input model parameters, the shear and longitudinal wave velocities of this model can be estimated to be 2435 m/s and 4551 m/s, respectively. According to the calculation equation of wavelength $\lambda = C_s/f$ ($\lambda = C_p/f$), where $C_s(C_p)$, $f$, and $\lambda$ represent the shear (longitudinal) wave velocity, highest waveform frequency, and wavelength, respectively. And the wavelength of this seismic wave can be estimated to be from 121 m to 228 m. Therefore, the zone size is set to 5 m for the perilous rock mass and 15 m for the base rock, satisfying the calculation requirements.

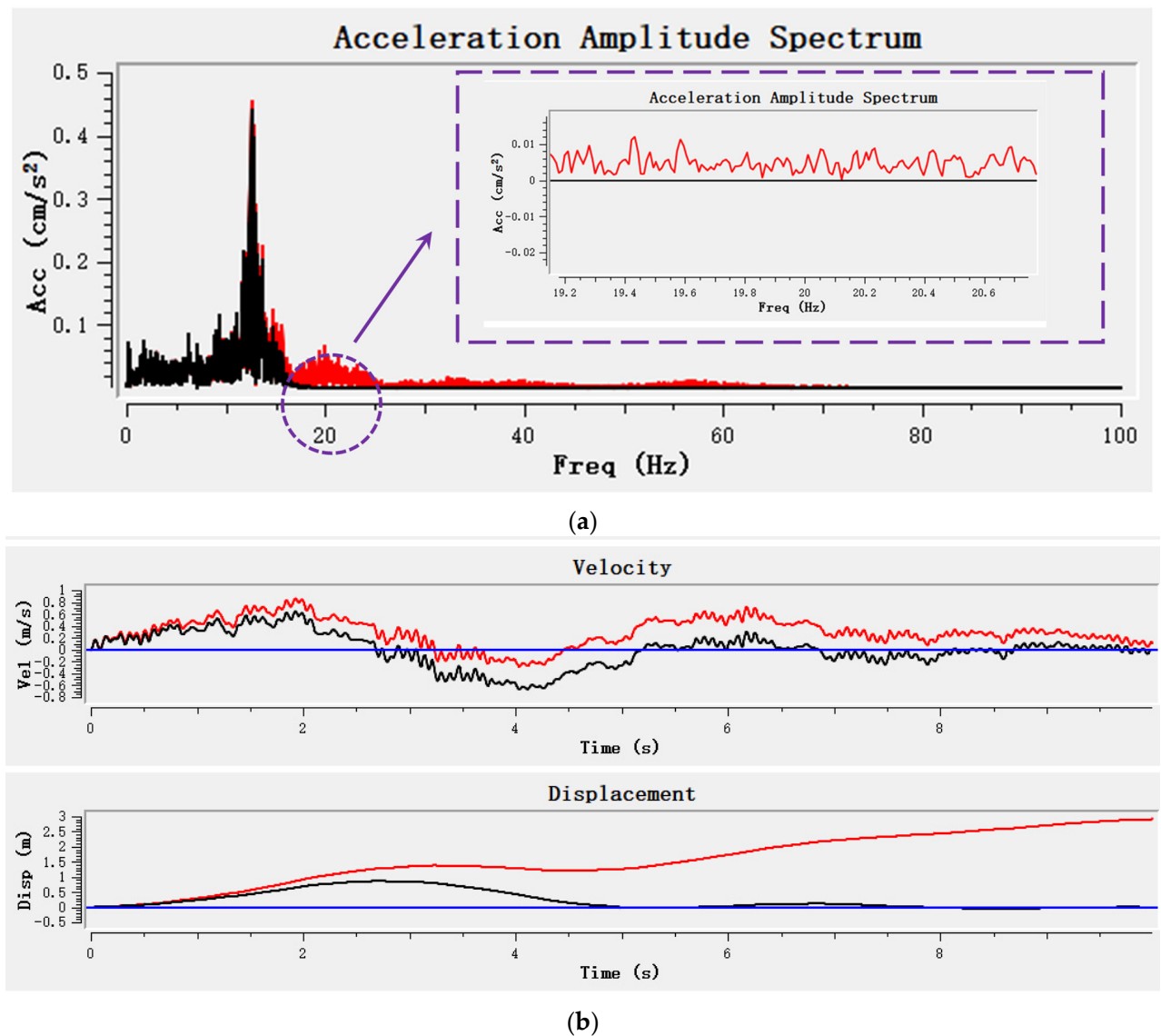

**(a)**

**(b)**

**Figure 9.** Seismic wave filtering and baseline correction. (**a**) Seismic wave filtering (black line: filtered waveform; red line: original waveform). (**b**) Baseline correction (black line: velocity and displacement curve after correction; red line: velocity and displacement curve before correction; blue line: reference line).

The velocity-time history within 45–55 s when the acceleration peak appears is intercepted as the input dynamic loading, and the bottom boundary of the model is the seismic application source. In the 3DEC seismic dynamic analysis, if the final input velocity is not zero, or the final displacement obtained by integrating the input velocity is not zero, the velocity and residual displacement will continue to appear at the end of the dynamic calculation. Baseline correction is usually conducted to eliminate the integration errors for time-domain seismic analysis by adding a low-frequency waveform to the original velocity time history. Figure 9b shows the baseline correction progress for EW seismic waves, and Figure 10 illustrates the final input of seismic waves of EW, NS, UD, and EW_NS_UD in 3DEC.

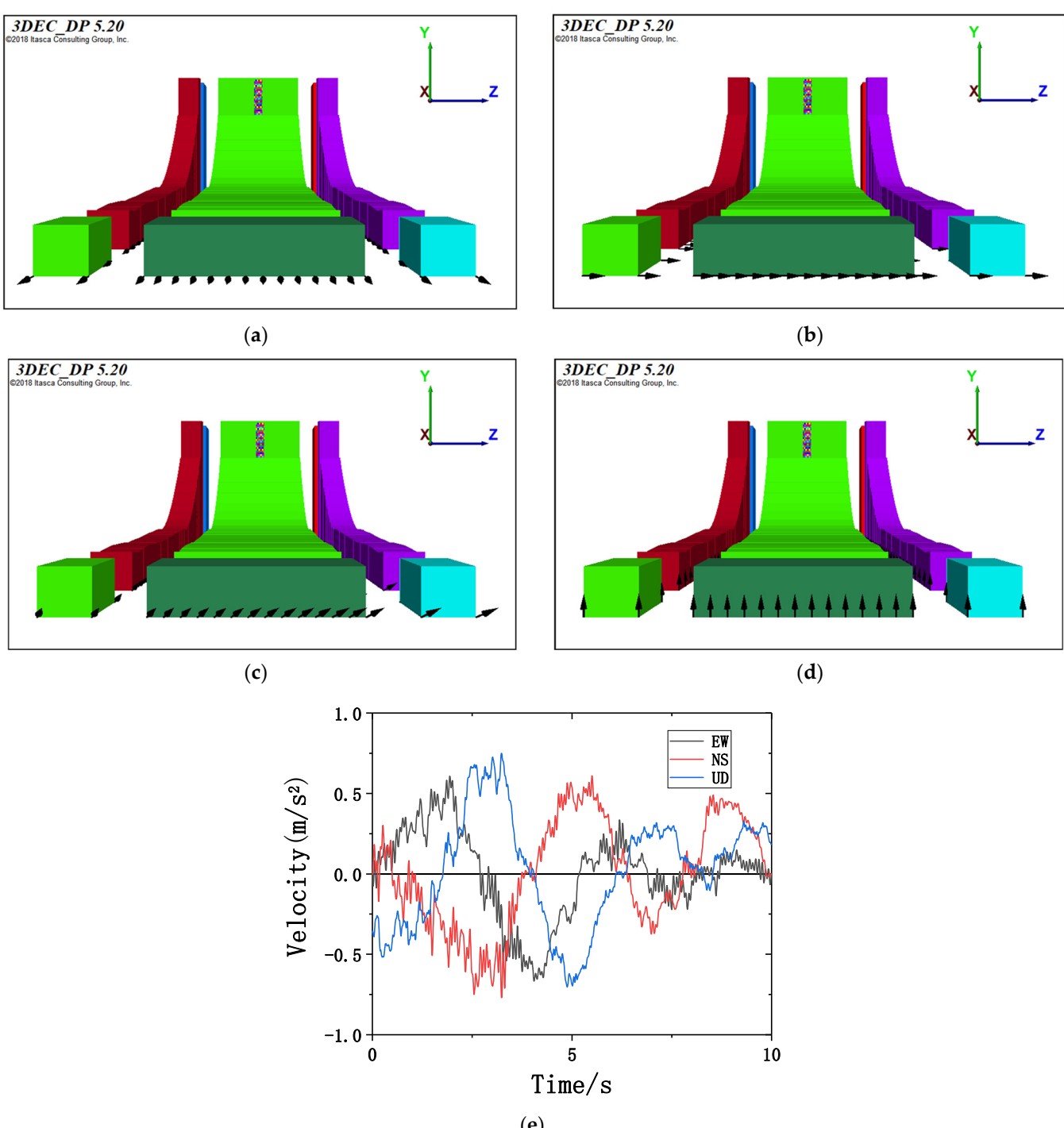

**Figure 10.** The input of seismic waves in 3DEC. (**a**) Case 1: EW seismic wave; (**b**) case 2: NS seismic wave; (**c**) case 3: UD seismic wave; (**d**) case 4: EW_NS_UD seismic wave. (**e**) The input seismic wave curves.

## 4. Results

### 4.1. Dynamic Response of the High-Steep Rock Slide

As an illustration, the velocity contour of the EW seismic wave propagation process is depicted in Figure 11. The numerical results show that the dynamic response of the slope does not occur at the same time. First, the velocity is generated at the model's bottom, as shown in Figure 11a, and then transmits from the bottom to the top. The propagation process has a strip-like characteristic: with the growing altitude, the velocity increases.

When the velocity field reaches the slope surface, the surface velocity is larger than the internal velocity at the same altitude, as shown in Figure 11b,c. It can be inferred that the input seismic waves are affected by vertical and surface amplification of the slopes. The other three cases have the similar dynamic response phenomenon.

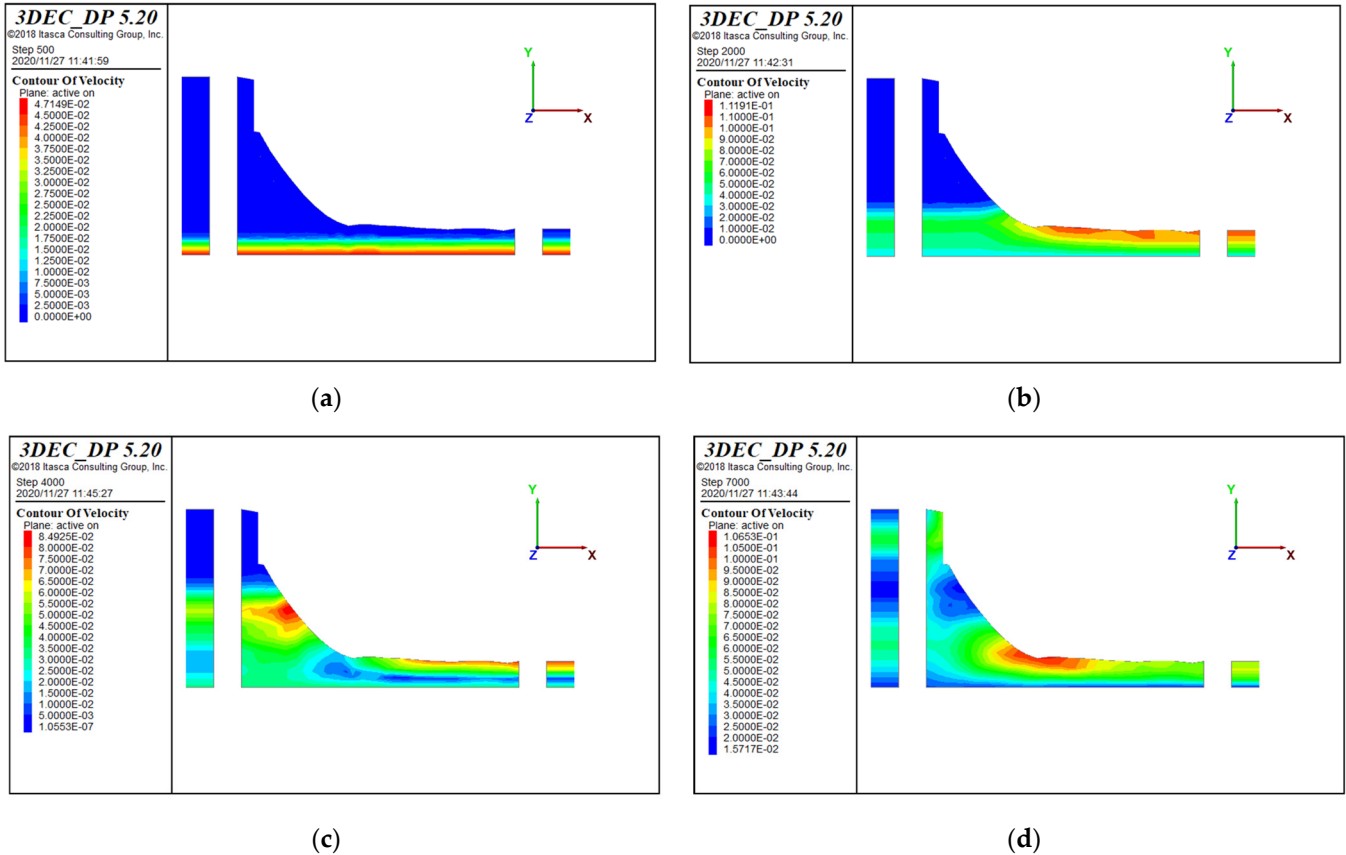

**Figure 11.** Dynamic response of the high-steep rock slide (under EW seismic wave loading): (**a**) 500 steps; (**b**) 2000 steps; (**c**) 4000 steps; and (**d**) 7000 steps.

Figure 11d shows the velocity field transmits to the top after 7000 steps and lasts about 0.13 s, equal to the calculation result. The dynamic response steps/time of the other three cases are NS: 8000/0.15 s, UD: 4000/0.07 s, and EW_NS_UD: 4000/0.07 s, also confirming the calculation results. The simulation results reveal that the seismic wave propagation time is affected by the type of seismic wave, and the propagation time of the EW_NS_UD seismic wave is consistent with the propagation time of the UD seismic single wave. We infer that the first velocity field reaching the top under EW_NS_UD seismic wave is the UD seismic wave, and the three seismic waves act independently. To verify this hypothesis, the velocity contour of UD wave, EW_NS_UD wave, and EW wave under 4000 steps are recorded, respectively. The velocity fields of monitor 1 and monitor 6 are selected for comparative analysis, as shown in Figure 12. Here, let $V_n$ denote the velocity of monitor n. It can be seen that the velocity of monitor 1/6 under the EW wave is both 0. It shows that under the EW single wave, the velocity field has not yet reached the position of monitor 1 and monitor 6. The velocities of monitor 1/6 under the UD seismic wave are 0.12 and 0.12 m/s, respectively. The velocities of monitor 1/6 under the EW_NS_UD wave are 0.13 and 0.14 m/s, respectively. Those two results are almost consistent. That is to say, at 0.07 s, the velocity fields generated by the UD wave and the EW_NS_UD wave at monitor 1 and monitor 6 are the same. The numerical results satisfy the hypothesis.

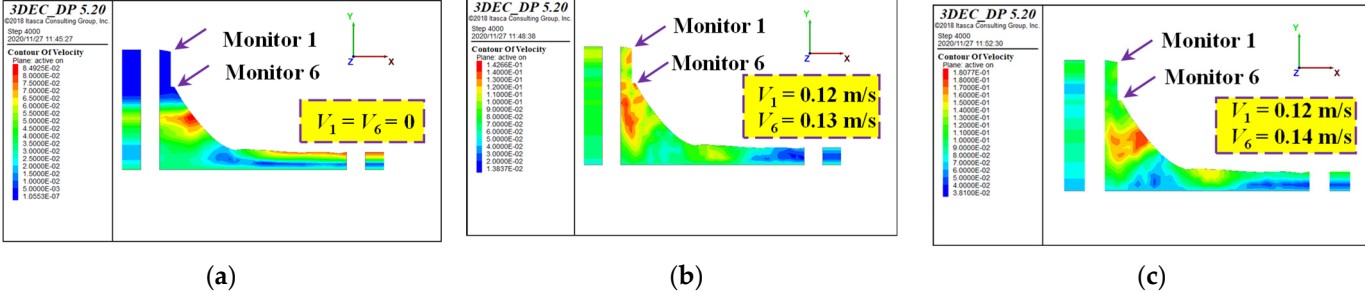

**Figure 12.** Dynamic response of the high-steep rock slide (4000 steps). (**a**) EW seismic wave; (**b**) UD seismic wave; and (**c**) EW_NS_UD seismic wave.

*4.2. Slope Failure Scenario of the High-Steep Rock Slide*

The failure behaviors of high-steep rock slide under cases 1/4 and cases 2/3 are different, which will be described separately below. Figure 13a,b show the post-failure behaviors of landslides in cases 1/4. The high-steep rock slide can be described by four-stage scenario.

(1) Stage 1 (peeling): In the time of 0–4/0–3 s, the front edge rock peels and collapses. At this stage, the front edge rock slides vertically. The rock mass collapses after it crashes the base rock.

(2) Stage 2 (toppling): In the time of 5–14/4–12 s, the rear rock cracks propagate, and then, the rock dumps. The crack propagation occurs on the sliding surface in contact with the base rock, and then, the rock begins to rotate with the bottom of the rock as the pivot. At the critical time 1, two vertical cracks develop inside the dangerous rock. At the critical time 2, the upper parts of the rear two blocks dump at the 1/3 height of the blocks. At the critical time 3, the front block breaks at the same position.

(3) Stage 3 (collapsing): In the time of 15–37/14–32 s, the perilous rock ejects and disintegrates. Stage 3 has two sub-stages: at the first sub-stage (15–21/14–17 s), more than 1/3 height rock mass crashes the slope, forming collapse debris. Then, it rolls down and leaves the bottom rock remaining. At the second sub-stage (21–37/17–32 s), with the complete dumping of the upper block, the lower block is gradually out of balance and then collapses at the critical time 4, causing the secondary collapse of the perilous rock mass.

(4) Stage 4 (depositing): This stage is the deposition process, and the convergence time is 79/82 s.

Figure 13c,d show the different failure behaviors of the rock slides in cases 2/3, which are not completely failing. The first stage is similar to that of cases 1/4, and both of the first stage times of NS/UD are 0–3 s. At the second stage, only the front crack propagates and dumps at the critical time 1 (NS/UD: 14/16 s). Compared to the dumping point with the cases 1/4, the cases 2/3 dumps at the bottom of rock mass, and the dumping phenomenon is not very apparent. Then, rock mass collides and accumulates until the system reaches equilibrium at 126/77 s, NS/UD.

Comparing the failure behavior of these four cases, it can be observed that EW and EW_NS_UD seismic waves have more vital damage to slope stability than NS and UD seismic waves. It can be concluded that the EW wave plays a dominant role in the failure behavior of landslides in three-directional seismic waves. Still, the existence of NS/UD seismic waves will accelerate the failure process of landslides.

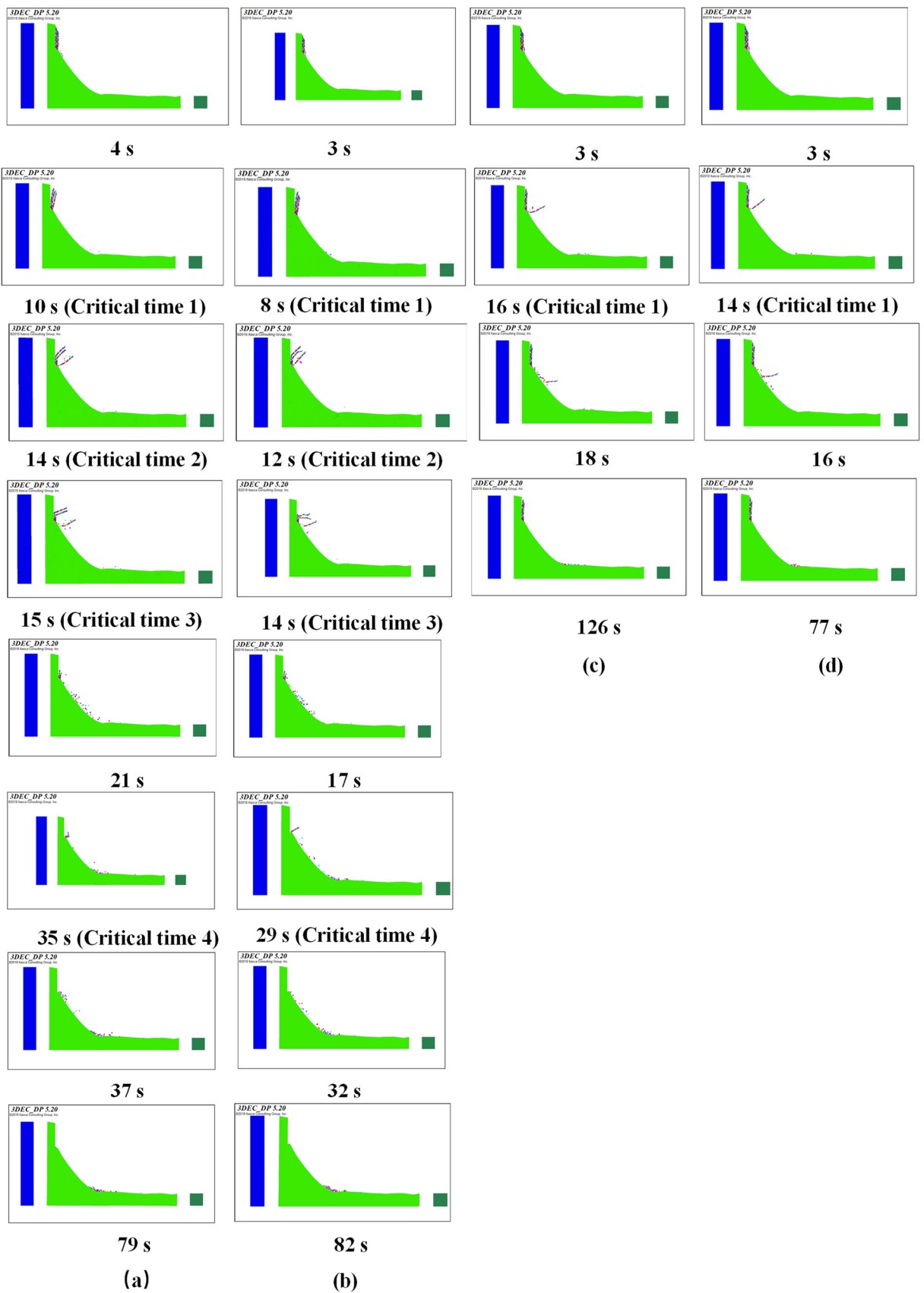

**Figure 13.** Failure behavior of the high-steep rock slide (**a**) case 1: EW seismic wave; (**b**) case 4: EW_NS_UD seismic wave; (**c**) case 2: NS seismic wave; and (**d**) case 3: UD seismic wave.

### 4.3. Analysis of the Velocities of the Monitors

Figure 14 shows the velocity-time history curves in four cases, and Figure 15 shows the comparison of maximum velocities of the monitors and their appearance time in these four cases. Here, let $V_n$ denote the velocity of monitor n ($n$ = 1–7). From the two horizontally arranged monitors 1/2/3 and monitors 4/7, we can observe that, in cases 2, 3, and 4, $V_3$ (dark blue line) is always greater than $V_2$ (red line), while it is approximately equal to $V_2$ in case 1. $V_1$ (black line) is the maximum value of the three monitors in case 1, while it is the minimum value in case 2. In conclusion, the velocity of the edge of the slope is more significant than that of the inside. $V_7$ (light blue line) is always 0 in cases 2/3. Therefore, only $V_4$ (green line) and $V_7$ in cases 1/4 are compared and analyzed. The result shows that $V_7$ is greater than $V_4$ in case 1, while $V_7$ and $V_4$ are approximately equal in case 4. We investigated that the velocity relationship between $V_4$ and $V_7$ was not apparent. The time for the max velocity appearance is that $V_3$ is earlier than $V_2$, earlier than $V_1$, and $V_4$ is earlier than $V_7$.

From the longitudinally arranged monitors 3/4/5/6 and monitors 1/7, it can be found that except for $V_4$ being more significant than $V_3$ in case 1, $V_3 > V_4 > V_5$ (purple line) in other cases. $V_1$ is more significant than $V_7$ in case 4, while it is the opposite in case 1. We conclude that the height of the landslide is positively correlated with the peak velocity. The velocity relationship between $V_6$ (yellow line) and the others is not apparent, which is just like $V_4$ and $V_7$, so we analyzed that the velocity of the sliding surface is not related to others. The time for the max velocity appearance is that $V_3$ is earlier than $V_4$, earlier than $V_5$, earlier than $V_6$, and $V_1$ is earlier than $V_7$.

### 4.4. Analysis of the Displacement of the Monitors

Figure 16 shows the displacement time history curves in four cases. The monitors without displacement are not shown in the figure. The number of effective monitors is 7/6/5/4 for four cases, respectively, and the displacement trend is similar. Here, let $D_n$ denote the displacement of monitor n ($n$ = 1–7).

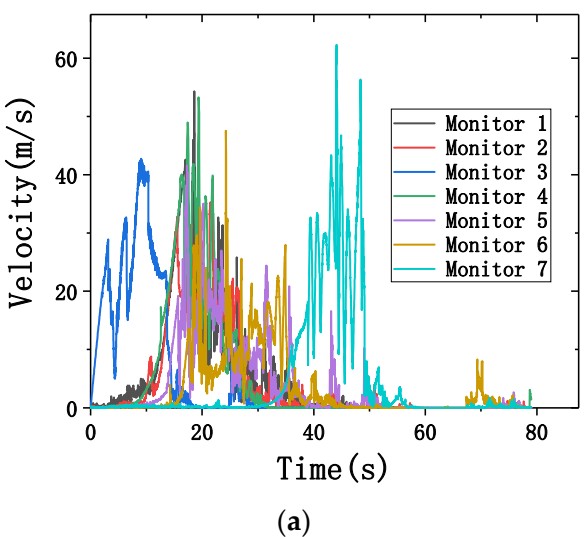

(**a**)

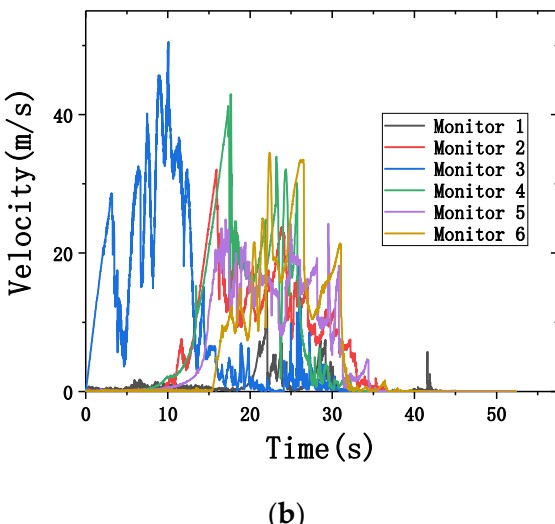

(**b**)

**Figure 14.** *Cont.*

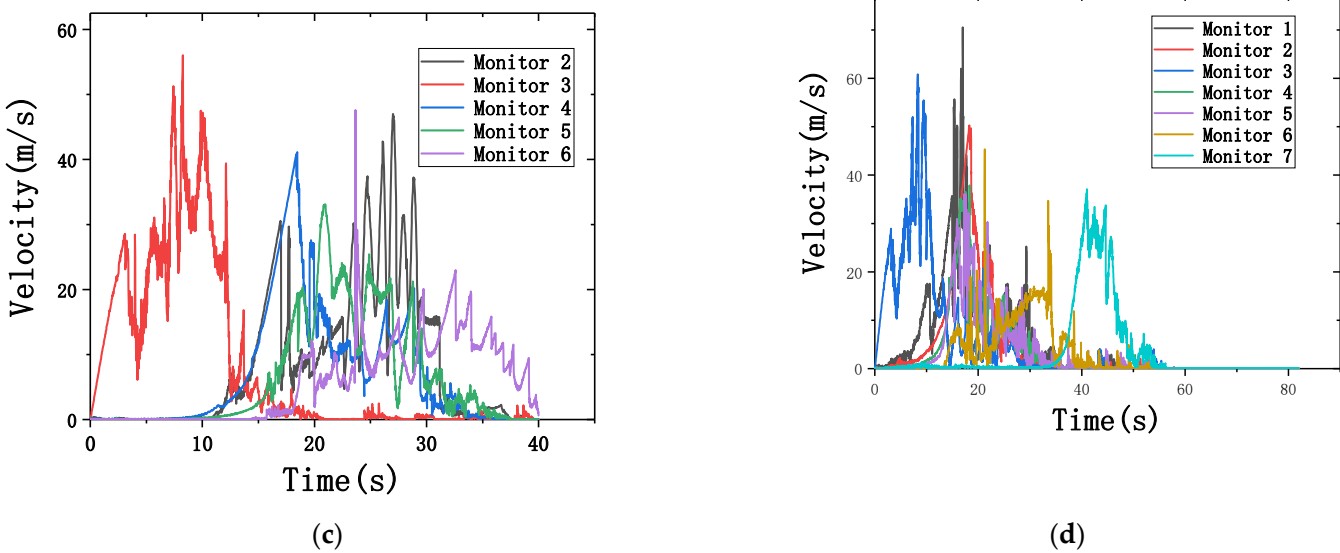

(**c**)                                      (**d**)

**Figure 14.** Time history curves of velocity of monitors in the high-steep rock slide process. (**a**) Case 1: EW seismic wave; (**b**) case 2: NS seismic wave; (**c**) case 3: UD seismic wave; and (**d**) case 4: EW_NS_UD seismic wave.

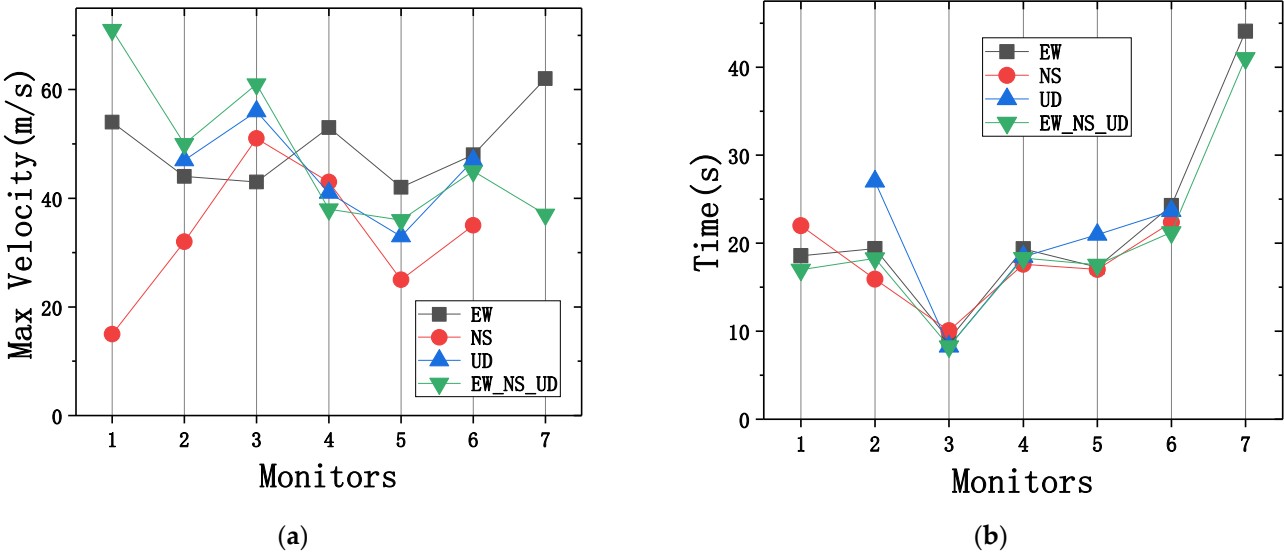

(**a**)                                      (**b**)

**Figure 15.** Comparison of maximum velocities of the monitors (**a**) and their appearance time (**b**) in 4 cases.

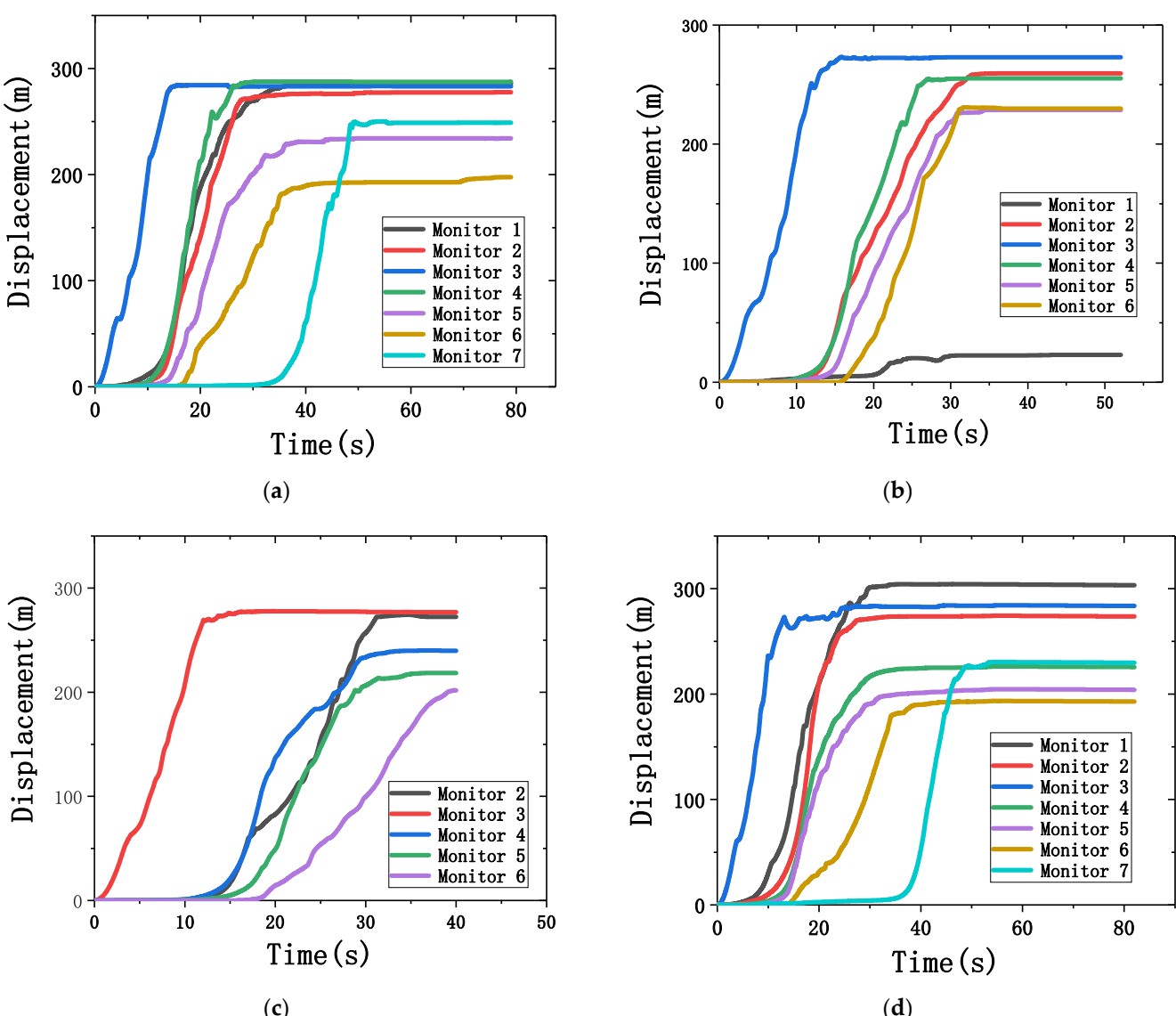

**Figure 16.** Time history curves of displacement of monitors in the high-steep rock slide process. (**a**) case 1: EW seismic wave; (**b**) case 2: NS seismic wave; (**c**) case 3: UD seismic wave; and (**d**) case 4: EW_NS_UD seismic wave.

As an illustration, Figure 17 shows the characteristics of case 1. From the two horizontally arranged monitors 1/2/3 (Figure 17a) and monitors 4/7 (Figure 17b), we can observe that the value of displacement is $D_3 > D_1 > D_2$ ($D_4 > D_7$), and the time for reaching the maximum displacement is that $D_3$ is earlier than $D_2$, which is earlier than $D_1$ ($D_4$ earlier than $D_7$). The results consist with the previous section conclusion, "the velocity of the slope's edge is larger than that of the inside". The results show that (1) the rear rock has a more significant displacement than the front rock, and the displacement of the edge of the slope is more significant than that of the inside; and (2) the stable time is inversely related to the distance from the edge rock. The other three cases also have the similar conclusion. It is worth mentioning that in case 4, $D_1$ is the largest, followed by $D_3$ and $D_2$.

From the longitudinally arranged monitors 3/4/5/6 (Figure 17c) and monitors 1/7 (Figure 17d), we can observe the displacement characteristics along the vertical direction. The order of displacement from large to small is $D_3 \approx D_4 > D_5 > D_6$ ($D_1 > D_7$), and the stable time is $D_3 < D_4 < D_5 < D_6$ ($D_1 < D_7$). The results show that the displacement values

are related to slope altitude under seismic loading, while stable time is inversely related, as is the case in the other three cases.

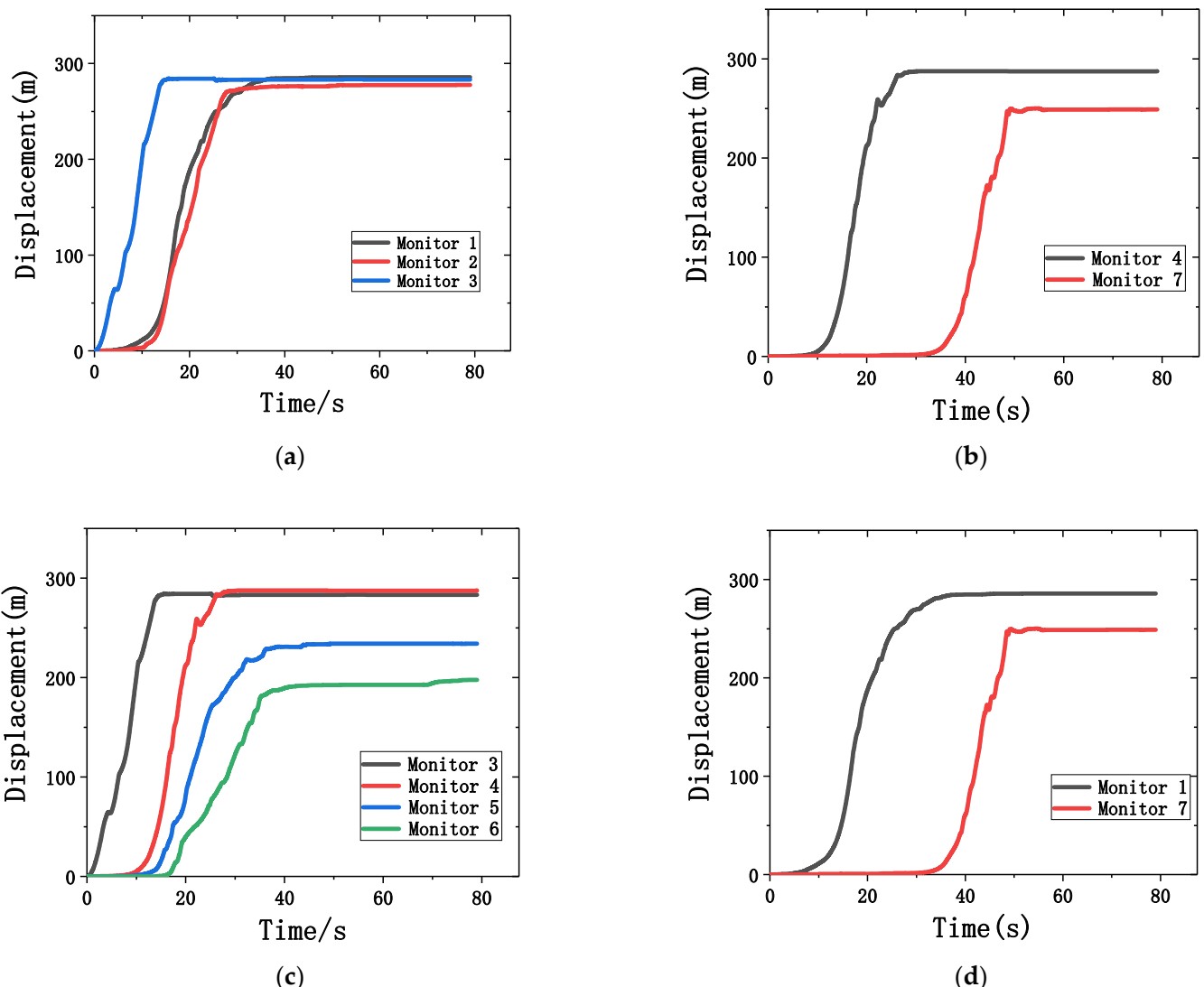

**Figure 17.** Time history curves of displacement of monitors under the EW seismic wave loading. Horizontally arranged monitors 1/2/3 (**a**) and 4/7 (**b**); longitudinally arranged monitors 3/4/5/6 (**c**) and 1/7 (**d**).

Figure 18a shows no significant difference in the displacement value of $D_3$ in four cases. The difference in displacement gradually increases as the altitude decreases. The order of $D_4$, which is case 1 > 2 > 3 > 4, has the largest displacement difference in different cases. $D_5$ is the same as $D_4$, and the displacement order is case 2 > 3 > 1 ≈ 4. In conclusion, the response of different seismic waves at low altitudes is different, and the effects of seismic waves in high-altitude areas are equivalent.

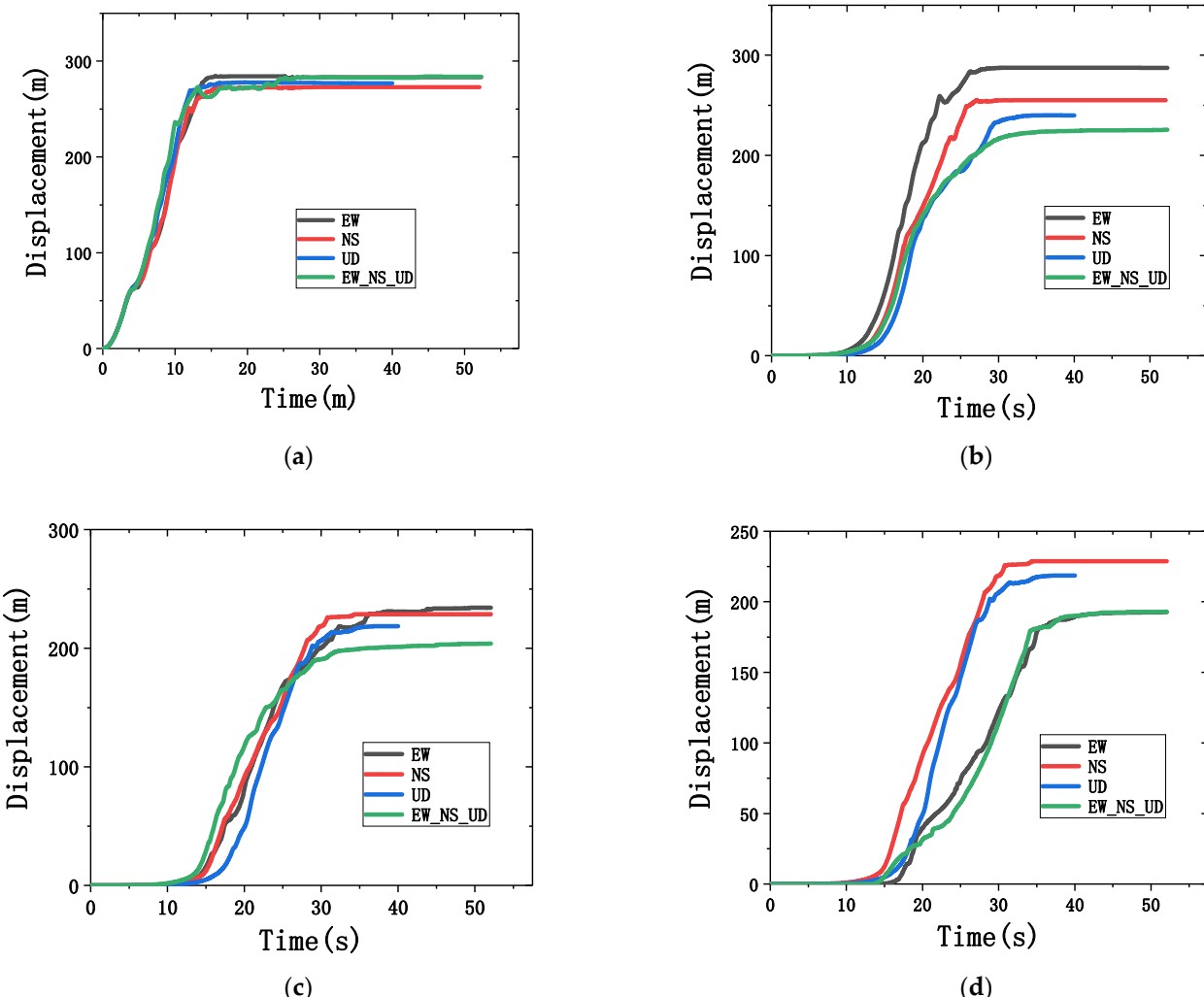

**Figure 18.** Time history curves of displacement of monitors 3/4/5/6 in four cases. (**a**) Monitor 3; (**b**) Monitor 4; (**c**) Monitor 5; and (**d**) Monitor 6.

*4.5. Analysis of the Characteristics of the High-Steep Rock Slide Deposition*

Figure 19 shows the rock slide deposition in four cases, and Table 2 presents the maximum displacement value (*X* direction and *Z* direction) and the main deposition area value. It shows that the deposition area of case 1 and case 4 are similar, which proves the view that the EW wave plays a dominant role in the three-directional seismic wave mentioned before. The farthest distance (*X* direction) of the landslide is in case 1, while in other cases, they are very close. It shows that the farthest landslide distance in the three-directional seismic wave is not necessarily more significant than others, for the possible reason that various waves may be superimposed or offset. In addition, the farthest distance (*Z* direction) in case 2 and the nearest is case 1. It shows that the displacement of the rock slide along the loading direction is the largest, while the vertical loading direction is the smallest.

**Table 2.** Deposition of high-steep rock slide in four cases.

| Cases | Farthest Distance (X Direction) | Farthest Distance (Z Direction) | Main Deposition Area (X Direction) |
|---|---|---|---|
| 1 EW | 450 m | −67–60 m | 160–240 m |
| 2 NS | 340 m | −89–60 m | 172–223 m |
| 3 UD | 360 m | −60–57 m | 176–221 m |
| 4 EW_NS_UD | 350 m | −74–60 m | 155–235 m |

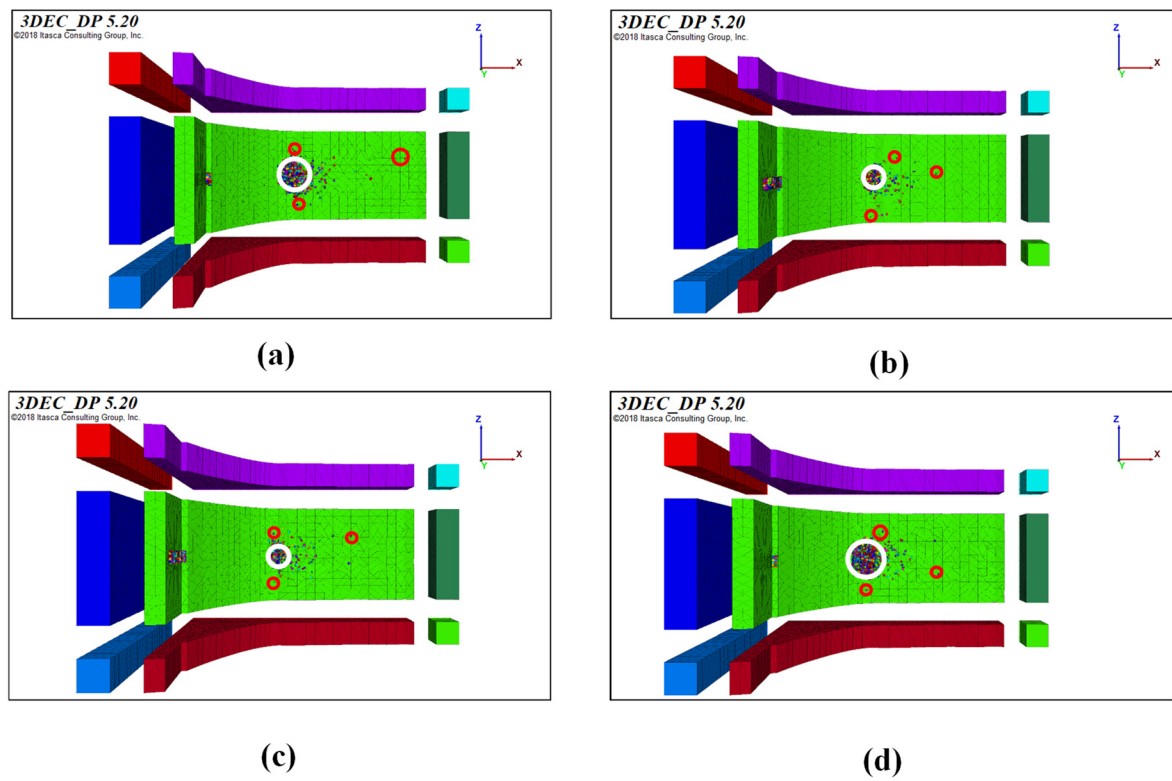

(a)　　　　　　　　　　　　　(b)

(c)　　　　　　　　　　　　　(d)

**Figure 19.** Deposition of high-steep rock slide in four cases (the three red circles in the figure represent the most distal, leftmost, and rightmost blocks, respectively, and the white circles are the main rock slide deposition). (**a**) Case 1: EW seismic wave; (**b**) case 2: NS seismic wave; (**c**) case 3: UD seismic wave; and (**d**) case 4: EW_NS_UD seismic wave.

*4.6. Impact of the Local Damping Ratio*

The local damping in this study is set to 0.005, which is obtained from many repeated trials. As mentioned in Section 3.2, local damping is used to reduce unnecessary numerical vibrations. If the local damping is too large, the rock slide will remain stable under all types of seismic waves. If the local damping is too small, the rock slide will float in the air, which is obviously unrealistic.

In the local damping ratio sensitivity analysis, a three-directional seismic wave is adopted. The value of damping ratio is conducted with 0.001, 0.005, and 0.01. The stability of the perilous rock mass increases with the growing value of local damping ratio (Figure 20). When the value of the local damping is 0.001, the rock slide will float in the air, and the 3DEC computations will diverge (Figure 20a). When the value of the local damping is 0.01, the rock slide is not completely failing (Figure 20b).

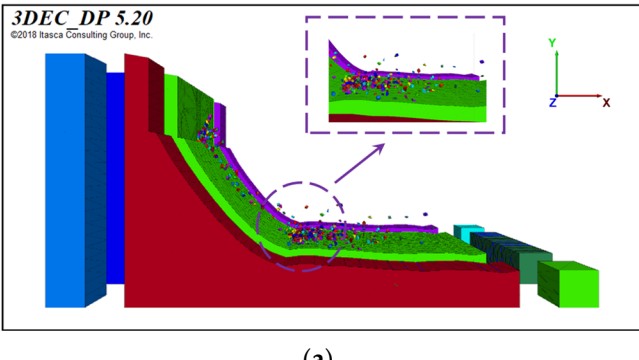 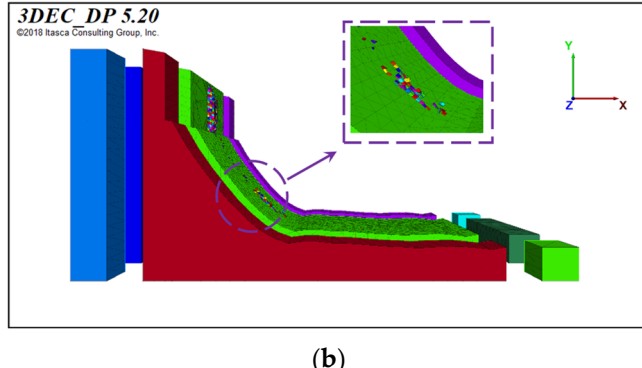

(**a**)　　　　　　　　　　　　　　　　　　　　　　　　(**b**)

**Figure 20.** Numerical results of local damping 0.001 (**a**) and 0.01 (**b**) under three-directional seismic wave.

### 4.7. Impact of the Dip Angle of Joints

The dip angle of joints might influence the failure behavior of the rock slide. In the joint dip angle sensitivity analysis, three-directional seismic wave is adopted. The dip angle in this study is conducted with 0°, 20°, and 60°. The failure behavior of joint dip angle 20° is shown as Figure 13b. Figure 21 shows the critical failure behavior of joint dip angle 0° and 60°. The failure behavior of the rock slide changes from toppling movement to sliding movement with the increasing joint dip angle.

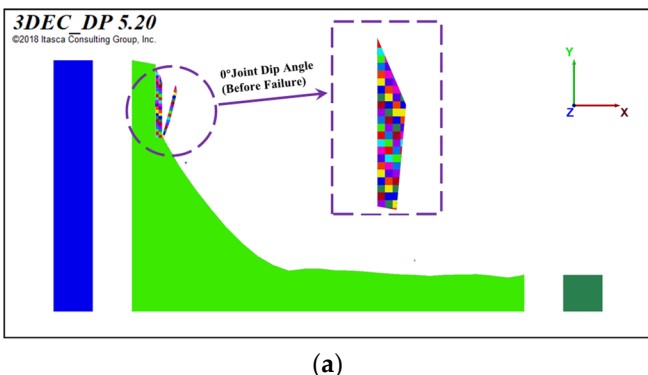 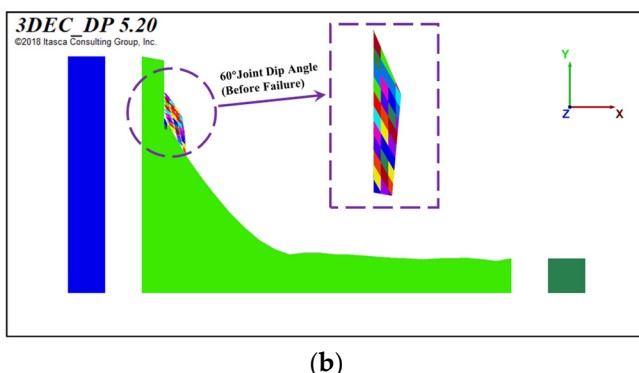

(**a**)　　　　　　　　　　　　　　　　　　　　　　　　(**b**)

**Figure 21.** Failure behavior of the rock slide with joints dip angle 0° (**a**) and 60° (**b**) under three-directional seismic wave.

### 5. Discussions

This study evaluates the dynamic response of a typical type of landslides, that is, high-steep rock slide, under different types of seismic waves. Part of the analysis of the numerical results have been discussed in previous section, and they will be described as conclusions in next section as well. In addition, there are still some discussions worth of attention.

(1) Selection of the input seismic waves:

At time of the Jiuzhaigou earthquake, China Earthquake Networks Center collected the seismic waves of 66 stations. The seismic waves selection criteria in this study are as follows: the station as close to the study area as possible. However, the recorded seismic waves are quite different in the total 66 stations, e.g., the amplitude. Therefore, it is worth discussing whether it is available to select seismic waves only by the value of distance.

(2) The validity of the typical high-steep rock slide numerical modeling:

This study is to explore the regularity of the dynamic response of the typical high-steep rock slide under different seismic waves. Therefore, the numerical model is simplified based on the actual landslide. However, the conclusions investigated in this study are

consistent with those obtained in other studies. For example, seismic waves have edge and altitude amplification effects [36], and the Up-Down seismic waves have little impact on the dynamic response of the landslide [29]. The available of the numerical results can be verified. The follow-up work will study a specific landslide to explore how to select seismic wave loading in 3DEC numerical modeling.

(3) Effects of the orientation of the slope:

Firstly, it should be noted that the free-field boundary in 3DEC requires that the all five surfaces (except the top surface) of the numerical model should be parallel to the coordinate axis. Moreover, the seismic wave loading is limited to the coordinate axis; that is, it can only be loaded along *X*, *Y*, and *Z* directions. Owing to the left-handed coordinate system adopted in the numerical modeling, the Up-Down seismic wave is determined along the *Y* direction. The horizontal seismic waves (East-West and North-South seismic waves) are corresponding to the *X* or *Z* directions. However, the actual orientation of the slope is not exactly towards north, south, east, or west. It may lead to the inconsistency of the seismic wave loading in numerical modeling with the actual vibration of the slope. For example, this study assumed that the orientation of the slope is towards east, corresponding to the *X* direction. Therefore, the input X-directional seismic wave loading can be regarded as the East-West seismic wave. Once the actual slope is not towards east, the *X* direction is not actually the East-West seismic wave.

(4) The reason for the numerical simulation time longer than that of the actual situation:

In general, the failure process of the rock slide occurs very fast and lasts for a few seconds. However, the convergence time of the numerical simulation is much longer than that of actual rock slide. The reason for the numerical simulation time longer than that of the actual situation might be the 3DEC calculation principle. The calculating unit of 3DEC is called "block". The overall system can be influenced by each single "block". Therefore, it may take a relatively longer time to ensure each "block" is in a stable state. However, in the general failure process of the landslide, the time of the failure process of the rock slide can be estimated roughly by observing the rock slide stops.

## 6. Conclusions

The effects of the four types of the seismic waves on the dynamic response and failure behavior of landslides were numerically investigated. The following conclusions were drawn:

(1) The velocity contour shows that the velocity increases with the growing altitude, and the surface velocity is more significant than the internal velocity at the same altitude. In other words, the dynamic response of the slope is affected by vertical and surface amplification. By contrast to the velocity field of East-West, Up-Down, and three-directional seismic waves, it can be investigated that each seismic wave actually affects the landslide independently and will not interfere with each other.

(2) By comparing the failure behavior of four cases, the East-West and three-directional seismic wave have more vital damages to slope stability than North-South and Up-Down seismic wave. The East-West wave plays a dominant role in the failure behavior of landslides in the three-directional seismic wave. Still, the existence of North-South and Up-Down seismic wave will accelerate the failure process of landslides.

(3) The displacement/velocity of the slope's edge is more significant than that of its inside, and the height of the landslide is positively correlated with the displacement/velocity. As for the distinctive effects of different waves, the effects of seismic waves on the displacement of landslides are different at low altitude, while the effects of seismic waves are equivalent at high altitude.

(4) The farthest distance (*X* direction) of the rock slide is in East-West seismic wave case, while in other cases, they are very close. The reason the earthquake can trigger long-distance movement of landslide is that the slope can obtain energy from the continuous shaking of the ground surface and convert this energy into kinetic energy, which is reflected

in the block movement. In addition, the displacement of the landslide along the dynamic loading direction is the largest, while the vertical loading direction is the smallest.

(5) The local damping ratio and the joints dip angle can affect the failure behavior of the rock slide. The stability of the perilous rock mass increases with the growing value of local damping ratio. The failure behavior of the rock slide changes from toppling movement to sliding movement with the increasing joint dip angle.

**Author Contributions:** Conceptualization, Z.G. and H.L.; Funding acquisition, Z.G and H.L.; Writing—original draft, Z.G. and H.L.; Writing—review & editing, Z.G and H.L. All authors have read and agreed to the published version of the manuscript.

**Funding:** This research was funded by the National Key Research and Development Plan of China (Grant No.: 2019YFC1509701).

**Institutional Review Board Statement:** Not applicable.

**Informed Consent Statement:** Not applicable.

**Data Availability Statement:** The data presented in this study are available on request from the corresponding author.

**Acknowledgments:** The authors would like to thank the editor and the reviewers for their helpful comments. Acknowledgement for the data support from "China Earthquake Networks Center, National Earthquake Data Center (http://data.earthquake.cn (accessed on 20 March 2020))".

**Conflicts of Interest:** The authors declare no conflict of interest.

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
