# Peer review of "Effects of Three-Directional Seismic Wave on Dynamic Response and Failure Behavior of High-Steep Rock Slide"

_applsci, doi:10.3390/app12010020_

Round 1
Reviewer 1 Report
The paper shows a numerical analysis of earthquake-induced landslide, focused on high-steep rock slopes and aimed at investigating the effects of the input motion, both in terms of dynamic properties and direction. The abstract and the introduction are overall clear, but the rest of the paper needs important improvements to make the discussion clearer and impressing. See my specific comments in the following.
- Line 27: For a wider view of the topic, the general effects of earthquakes (even on buildings) should be mentioned, together with the attraction of the Geotechnical community on the mitigation of these phenomena through interventions into the soil.
See:
- https://doi.org/10.1080/13632469.2020.1779871
- https://doi.org/10.1080/13632469.2021.1961933
- https://doi.org/10.3390/geosciences11050201
- The high-steep slope shown in Fig. 2a is described as "an example" in Section 2.2 (Line 119); however, the reader "discovers" that this "example" is exactly the slope under examination only in Section 3.3. It would be better to provide a clearer and detailed of the case study, including a discussion on how geometry (Section 3.3) and parameter (Table 1) have been derived.
- The title of Section 3.2 is not appropriate because the authors do not provide a description of DEM (as they have mentioned) but only some details of 3DEC, i.e. the code they used for the modeling.
- Eq. (1): Velocity are lowercase in the expression, but uppercase in the text. Please check it.
- Line 176: Vxxff, Vyyff and Vzzff are described as "forces", but this seems to be a mistake: it is physically impossible to subtract forces from velocities in Eq. (1)! Please check it
- Figure 6 is not well described in the text. What are the illustrated dampers?
- Line 244: Figure 9a does not show the mentioned "EW Seismic wave" but its Fourier spectrum. Please check it
- Figure 9: Subtitle "a)" misses
- Lines 247-248 refer to wave velocity as Cp and Cs; then, in Line 251 (Wavelength evaluation) velocity is indicated as "u". Please, make your formulation coherent.
- Have the numerical analyses been performed in time or frequency domain?
Author Response
Cover Letter for Revised Submission
December 16, 2021
Dear Editor and Reviewers,
We would like to submit our paper entitled “Effects of Three-directional Seismic Wave on Dynamic Response and Failure Behavior of High-Steep Rock Slide” for your consideration for publication in the journal Applied Sciences.
We have made a point-by-point response to the reviewers’ comments and suggestions, including a detailed description of any requested or suggested revisions.
We have also carefully checked and corrected the writing format and errors to make our revised manuscript conform to the journal style.
All the modifications and explanations in this revised version are listed in detail in the following “Responses to Reviewer's Comments”.
We would deeply appreciate your consideration and reviewers’ helpful comments and suggestions.
Yours Sincerely,
Ziwei Ge, Hongyan Liu*
School of Engineering and Technology, China University of Geosciences (Beijing)
BGI Engineering Consultants LTD, Beijing, China;
Email: lhy1204@cugb.edu.cn (HY. Liu)

Reviewer 2 Report
- Figure 5, it is better to show 3 steps with 3 different colors, to make the flowchart more visible.
- Line 197, it is better to explain for the readers in the text what is strata.
- In line 207, the authors need to explain why they considered angles 0 20 and 90º.
Author Response

(The authors gave the same response as above.)

Reviewer 3 Report
This article presents a study on the influence of different seismic vibration components on landslides, which is a subject of great interest.
There are 41 references in the article. About 16% are from the last 5 years, and about 40% are over 10 years old, which seems to me a little outdated. This is a bit strange, as I have come across several very recent papers on earthquake landslide evaluation models, such as the following examples, all from 2021:
https://doi.org/10.1016/j.enggeo.2021.106412
https://doi.org/10.1016/j.enggeo.2021.106177
https://doi.org/10.1016/j.enggeo.2021.106477
There are also recent papers about the Jiuzhaigou region, such as:
https://doi.org/10.1016/j.catena.2020.104851
The abstract seems a bit long. I counted 256 words, which slightly exceeds the maximum value of 200 that is stipulated in the journal's rules.
I consider the subject under study of interest for publication.
However, I have some concerns about the presented study.
My first concern is about the damping ratio, namely the value presented in line 187 of 0.5% (0.005). The authors should better explain how this value was obtained. What trials were carried out? And the results were compared against what?
Another concern is about the numerical model. It is well known that the results obtained with a distinct element method (DEM) are sensitive to the adopted discretization, namely the inclination of the faces of the discrete elements. This is particularly important because authors stated in lines 221-222 that all blocks and joints follow the linearly elastic model and Coulomb-slip model, respectively. So, this probably means that sliding, in the non-linear domain, will only occur in the contacts between adjacent blocks. For this reason, the authors must present a better image of the model with the representation of the direction of the block sliding faces (it is not very clear in Figure 7).
Moreover, a sensitivity analysis should be presented considering different meshes, which also might influence the failure pattern. For example, the rotation that is observed in Figure 13 might be changed to a sliding movement instead, and probably not such low damping is needed in order to assure the failure. In my opinion, this should be tested, because it might influence the results and conclusions. For example, If the block elements present horizontal faces, then the vertical component is not important. If discrete elements have very steep faces, then the vertical component gains more importance. In my opinion, it is mandatory to clarify this issue.
For these reasons, I recommend a major revision of the paper before publishing it.
Author Response

(The authors gave the same response as above.)

Round 2
Reviewer 1 Report
My suggestions have been properly addressed.
The manuscript quality has been greatly improved and the paper is now ready for publication.
Reviewer 3 Report
The authors have addressed all my previous concerns. They updated the literature review and carried out the sensitivity studies that I recommended. The results are what I expected to be, so I recommend publishing the paper.
This manuscript is a resubmission of an earlier submission. The following is a list of the peer review reports and author responses from that submission.